# Data Poisoning Attacks on Off-Policy Policy Evaluation Methods

**Elita Lobo**[1]        **Harvineet Singh**[2]        **Marek Petrik**[1]        **Cynthia Rudin**[3]        **Himabindu Lakkaraju**[4]

[1]University of New Hampshire, Durham, NH, USA
[2]New York University, New York, NY, USA
[3]Duke University, Durham, NC, USA
[4]Harvard University, Boston, MA, USA

## Abstract

Off-policy Evaluation (OPE) methods are a crucial tool for evaluating policies in high-stakes domains such as healthcare, where exploration is often infeasible, unethical, or expensive. However, the extent to which such methods can be trusted under adversarial threats to data quality is largely unexplored. In this work, we make the first attempt at investigating the sensitivity of OPE methods to marginal adversarial perturbations to the data. We design a generic data poisoning attack framework leveraging influence functions from robust statistics to carefully construct perturbations that maximize error in the policy value estimates. We carry out extensive experimentation with multiple healthcare and control datasets. Our results demonstrate that many existing OPE methods are highly prone to generating value estimates with large errors when subject to data poisoning attacks, even for small adversarial perturbations. These findings question the reliability of policy values derived using OPE methods and motivate the need for developing OPE methods that are statistically robust to train-time data poisoning attacks.

## 1 INTRODUCTION

In reinforcement learning (RL), off-policy evaluation (OPE) methods are popularly used to estimate the value of a policy from previously collected data [Thomas et al., 2015, Voloshin et al., 2020, Levine et al., 2020]. These methods are instrumental in high-stakes decision problems such as in medicine and finance, where deploying a policy directly is often infeasible, unethical, or expensive [Gottesman et al., 2020, Ernst et al., 2006b]. In such cases, one must estimate the value solely from a batch of data collected using a different and possibly unknown policy. Only if the OPE methods

estimate the value of a policy to be sufficiently high will stakeholders deploy it. Otherwise, the policy will be rejected. It is therefore essential that OPE methods do not severely overestimate the values of bad policies or underestimate the values of good policies [Gottesman et al., 2020].

Despite the importance of OPE methods, their sensitivity to adversarial contamination of logged data is not well understood. The complexity of OPE methods offers ample opportunities for attackers to introduce significant errors in OPE estimates with only small perturbations to the input data. For example, some OPE methods compute the value of a policy in a given state as a function of its value in future states. Therefore, even small errors introduced in the value estimates of these future states can accumulate and result in significant errors in the value estimates at the initial states, where critical strategic decisions are often made. This property could be exploited by attackers. Another possible avenue for an attack is the *importance sampling weights*. Popular OPE methods, such as the Doubly Robust and the Importance Sampling methods [Jiang and Li, 2016, Voloshin et al., 2020] use importance sampling weights to correct for dataset mismatch when evaluating the given policy with logged data from a different policy. The weights depend on the estimate of the policy used for the logged data. Attackers could perturb the data in a way that forces the agent to wrongly estimate the policy used to collect data and consequently introduce significant errors in the value estimates. Such vulnerabilities motivate the need for a thorough analysis of the effect of data poisoning attacks on OPE methods.

While some prior works have studied adversarial attacks in the context of policy learning in online and batch RL settings [Rakhsha et al., 2020, Ma et al., 2019, Chen et al., 2019], they mainly focus on teaching an agent to learn an adversarial policy or driving the agent to an adversarial state [Rakhsha et al., 2020, Zhang et al., 2021], and do not specifically investigate the effect of these attacks on OPE methods. In this work, we address the aforementioned gaps and study the effect of data poisoning attacks on OPE

*Accepted for the 38th Conference on Uncertainty in Artificial Intelligence* (UAI 2022).

methods. More specifically, our work answers the following question: *Can we construct small perturbations to the training data that significantly change a given OPE method's estimate of the value of a given policy?* To this end, we propose a novel data poisoning framework to analyze the sensitivity of model-free OPE methods to adversarial data contamination at train time. We formulate the data poisoning problem as a bi-level optimization problem and show that it can be adapted to diverse model-free OPE methods, namely, Bellman Residual Minimization (BRM) [Farahmand et al., 2008], Weighted Importance Sampling (WIS), Weighted Per-Decision Importance Sampling (PDIS) [Precup, 2000, Powell and Swann, 1966, Rubinstein, 1981], Consistent Per-Decision Importance Sampling (CPDIS) [Thomas, 2015], and Weighted Doubly Robust methods (WDR) [Jiang and Li, 2016]. To solve the aforementioned bilevel optimization problem in a computationally tractable manner, we derive an approximate algorithm using influence functions from robust statistics [Koh et al., 2018, Koh and Liang, 2017, Diakonikolas and Kane, 2019, Broderick et al., 2021]. To the best of our knowledge, our work is the first to study the sensitivity of a wide range of OPE methods to train-time data poisoning attacks.

We evaluate our framework using five different datasets spanning medical (Cancer and HIV) and control (Mountain Car, Cartpole, Continuous Gridworld) domains. Our experiments show that corrupting only 3%–5% of the observed states achieves more than $340\%$ and $100\%$ error in the estimate of the value function of the optimal policy in the HIV and MountainCar domains, respectively. Through our experimental results, we show that out of the five OPE methods, WDR, PDIS, and BRM are generally the least statistically robust, and CPDIS and WIS are relatively more statistically robust to such adversarial contamination. Finally, our findings question the reliability of policy values derived using OPE methods and motivate the need for developing OPE methods that are statistically robust to train-time data poisoning attacks.

## 2 PRELIMINARIES

We model a sequential decision-making problem as a Markov Decision Process (MDP). An MDP is a tuple of the form $\langle \mathcal{S}, \mathcal{A}, R, P, p_0, \gamma \rangle$ representing the set of states, set of actions, reward function, transition probability model, initial state distribution, and discount factor respectively. When taking an action $a \in \mathcal{A}$ in a state $s \in \mathcal{S}$ and transitioning to state $s' \in \mathcal{S}$, the scalar $R(s, a, s')$ denotes the reward received by the agent and $P(s, a, s')$ denotes the probability of transitioning to a state $s'$ on taking an action $a$ in a state $s$.

A randomized policy $\pi : \mathcal{S} \to \Delta^{|\mathcal{A}|}$ prescribes the probability of taking each action in each state. The value function $v^\pi : \mathcal{S} \to \mathbb{R}$ of a policy $\pi$ at a state $s$ is the expected dis-

counted returns of the policy starting from state $s$ and is given by

$$
v^\pi(s) = \mathbb{E}\left[\sum_{t=0}^{\infty} \gamma^t R(S_t, A_t, S'_{t+1}) \mid \pi, S_0 = s\right].
$$

The value of a policy is computed as $p_0^T v^\pi$. The state-action value function (also known as the Q-value function) $q^\pi : \mathcal{S} \times \mathcal{A} \to \mathbb{R}$ of a policy $\pi$ at a state $s$ and an action $a$ is the expected discounted returns obtained by taking action $a$ in state $s$ and following policy $\pi$ thereafter. The state-action value function $q^\pi$ for a policy $\pi$ is the unique fixed point of the *Bellman operator* $\mathcal{T}^\pi : \mathcal{S} \times \mathcal{A} \to \mathbb{R}^{\mathcal{S} \times \mathcal{A}}$ defined as

$$
(\mathcal{T}^\pi q)(s, a) = \tag{1}
$$
$$
\sum_{s' \in \mathcal{S}} \sum_{a' \in \mathcal{A}} \left( R(s, a, s') + \gamma P(s, a, s')\pi(s', a')q(s', a') \right).
$$

We assume the standard batch RL setting (e.g. [Levine et al., 2020]) in which the agent is given a batch of $n = N \times T$ transition tuples $D = ((s_j^i, a_j^i, r_j^i)_{j=1}^T)_{i=1}^N$, generated by a behavior policy $\pi_b$ for $N$ episodes of length $T$. The *goal* of OPE is to use $D$ to evaluate the value of the evaluation policy $\pi$. Let $D_0$ be a set of initial states sampled from the distribution $p_0$.

The value function is approximated using features $\xi : \mathcal{S} \to \mathbb{R}^d$. As is standard in linear value function approximation, we assume also that the state-action value function $q^\pi$ is approximated as a linear combination of state-action features $\phi : \mathcal{S} \times \mathcal{A} \to \mathbb{R}^{|\mathcal{A}| \cdot d}$. The state-action features for a given state-action pair $(s, a)$ are constructed by using the state features $\xi(s)$ at the indices corresponding to $a$ and zero elsewhere, i.e. $\phi(s, a)[ad : (a+1)d] \leftarrow \xi(s)$. Because the value function is estimated from data, we need to define a sample feature matrix $\Phi \in \mathbb{R}^{n \times d}$ where the rows correspond to the state-action features $\phi(s, a)$ for the $n$ state-action pairs in $D$. Similarly, $\Phi_p \in \mathbb{R}^{n \times d}$ denotes the sample feature matrix for the *next states* such that each row corresponds to $\phi(s'_i, \pi(s'_i))$ for the next states $s'_i$ in $D$. The sample reward matrix $r \in \mathbb{R}^{n \times 1}$ is constructed such that the $i^{th}$ row corresponds to the reward $r_i$ in $D$. More details on the construction of the sample feature matrices $\Phi$, $\Phi_p$ and reward matrix $r$ can be found in Section 4 in [Lagoudakis and Parr, 2003].

OPE methods are broadly classified into three categories: Direct, Importance Sampling, and Hybrid Methods [Voloshin et al., 2020]. *Direct Methods* estimate the value of the evaluation policy by solving for the fixed point of the Bellman Equation (1) with an assumed model for the state-action value function $q$ or the transition model $P$. We illustrate our attack on one of the most popular Direct Methods, namely the *Bellman Residual Minimization* (BRM) method [Voloshin et al., 2020, Farahmand et al., 2008]. BRM solves a sequence of supervised learning problems

with state-action features $\phi(s, a)$ as the predictor and the 1-step Bellman update $\mathcal{T}^\pi q = r + \gamma P q$ as the target response. The objective optimized in BRM is the Mean Squared Bellman residual (MSBR), defined as a weighted $L_2$ norm:

$$\text{MSBR}(\eta) \ = \ \|q_\eta - \mathcal{T}^\pi q_\eta\|_W^2 \ . \tag{2}$$

Here, the linear Q-value function $q_\eta$ is parameterized by $\eta$ as $q = \Phi \eta$. The weight matrix is computed as $W = \text{diag}[\mu^\pi]$ where $\mu^\pi \in [0, 1]^S$ represents the stationary state distribution of policy $\pi$. The value of a policy is then computed as

$$v_{\text{BRM}} \ = \ \sum_{s \in D_0} \sum_{a \in \mathcal{A}} p_0(s) \cdot \pi(s, a) \cdot q_\eta(s, a) \ . \tag{3}$$

*Importance Sampling Methods* (IS) [Kahn and Marshall, 1953] are based on Monte-Carlo techniques and compute unbiased but high-variance value estimates. The key idea is to compute the value of policy $\pi$ as the weighted average of the returns of the trajectories in $D$, where each trajectory is re-weighted by its probability of being observed under evaluation policy $\pi$. We focus on attacking three popular variants of importance sampling methods, namely the *Per-Decision, Consistent Weighted Per-Decision*, and *Weighted* IS methods (PDIS, CPDIS, WIS) [Precup, 2000, Thomas, 2015, Rubinstein, 1981]. Let $g_T^i = \sum_{t=0}^T \gamma^t r_t^i$ represent the returns observed for the $i^{\text{th}}$ trajectory in the dataset $D$ and assume that the behavior policy is parameterized by $\theta_b$ and estimated from data $D$ using maximum likelihood estimation (MLE) [Vaart, 1998]. In this setting, the MLE method effectively minimizes the Cross Entropy Loss (CEL) on the predictions of the behavior policy. In order to define the OPE estimates of the value functions, we need the importance sampling weights $\rho_{0:t}^i$ for time step $t$ defined as

$$\rho_{0:t}^i = \prod_{t'=0}^t \frac{\pi(s_{t'}^i, a_{t'}^i)}{\pi_b^{\theta_b}(a_{t'}^i | s_{t'}^i)} \ .$$

Here, the estimate of the behavior policy is defined as $\pi_b^{\theta_b}(a|s) = \exp(\phi(s, a)\theta_b)(\sum_{a' \in \mathcal{A}} \exp(\phi(s, a')\theta_b))^{-1}$ for each $s \in \mathcal{S}$ and $a \in \mathcal{A}$. Then the WIS, PDIS, and CPDIS value function estimates are defined as

$$v_{\text{WIS}} = \left( \sum_{i=1}^N \rho_{0:T}^i \right)^{-1} \sum_{i=1}^N \rho_{0:T}^i g_T^i, \tag{4}$$

$$v_{\text{PDIS}} = \frac{1}{N} \sum_{i=1}^N \sum_{t=1}^T \gamma^{t-1} \rho_{0:t}^i r_t^i, \tag{5}$$

$$v_{\text{CPDIS}} = \sum_{t=1}^T \gamma^{t-1} \frac{\sum_{i=1}^N \rho_{0:t}^i r_t^i}{\sum_{i=1}^N \rho_{0:t}^i}. \tag{6}$$

*Hybrid Methods* combine both Direct and IS methods to generate value estimates with low bias and variance. An important hybrid method is the *Doubly Robust* (DR) estimator [Jiang and Li, 2016], which decreases the variance in the

IS estimate by using the estimate from a method like BRM. The DR and Weighted DR (WDR) estimators are given by

$$v_{\text{DR}} = \frac{1}{N} \sum_{i=1}^N \sum_{t=0}^{T-1} \rho_{0:t}^i w_t^i + \frac{1}{N} \sum_{i=1}^N v_\eta(s_0^i).$$

$$v_{\text{WDR}} = \sum_{i=1}^N \sum_{t=0}^{T-1} \frac{\rho_{0:t}^i}{\sum_{i=1}^N \rho_{0:t}^i} w_t^i + \frac{1}{N} \sum_{i=1}^N v_\eta(s_0^i). \tag{7}$$

where $w_t^i \ = \ (r_t^i - q_\eta(s_t^i, a_t^i) + v_\eta(s_t^i))$ and $v_\eta(s_t^i) = \sum_{a \in \mathcal{A}} \pi(s_t^i, a) \cdot q_\eta(s_t^i, a)$. Here the parameters of the value function $q$ are estimated using Direct Methods like BRM. Because empirical studies show that there are no clear winners among the three methods [Voloshin et al., 2020], we investigate attacks on representative methods from each type.

# 3 DOPE FRAMEWORK

We first present our attack framework called DOPE for *D*ata poisoning attacks on *O*ff-*P*olicy *E*valuation. Then we demonstrate how to use the framework to attack the three types of OPE methods discussed in Section 2. The objective and scope of the attacks considered in DOPE are as follows.

**Scope**: We assume the setting of a white-box attack, i.e. the attacker has access to the batch data $D$, evaluation policy $\pi$, the value of the discount factor $\gamma$, and the attacker knows how the agent estimates the behavior policy and the state-action value function from the data. This kind of a setting is commonplace in the healthcare domains [Gottesman et al., 2020, Ernst et al., 2006a, Yu et al., 2021] where models are typically benchmarked and often made available to the general public so that they can be independently vetted and validated before deployment. Further, for the attack to be unnoticeable, we allow the attacker to only perturb at most $\alpha$ fraction of the transitions in $D$ while conforming to some perturbation budget $\varepsilon \geq 0$ to be defined later.

**Objective**: The goal of the attacker is to add small adversarial perturbations to a subset of transitions in $D$ such that it maximizes the error in the value estimate of a given policy in the desired direction. This means that the attacker may choose to decrease or increase its estimated value for the policy being evaluated such that a good evaluation policy is rejected or a bad evaluation policy is approved.

**Components:** The DOPE framework for a given OPE method has four major components: *Features ($\Psi$):* the part of the transition tuples targeted by the attack; *Value estimation function ($\rho$):* function used by the OPE method for computing the value; *Estimated parameter ($\theta$):* model parameters learned by the OPE method from the data; *Loss function (L):* loss optimized by the OPE method for model-fitting. We define each component in detail in Section 3.1. We can now formulate our attack model as a problem of

finding the perturbation matrix $\Delta = (\delta_i)_{i=1}^n, \delta_i \in \mathbb{R}^Q$ that maximizes the difference between values found using the perturbed and the original data under constraints dictating that the perturbations are small:

$$\underset{\Delta \in \mathbb{R}^{n \times Q}}{\text{maximize}} \quad \rho(\theta_{\text{pert}}, \Psi + \Delta) - \rho(\theta_{\text{org}}, \Psi) \tag{8a}$$

$$\text{subject to} \quad \theta_{\text{pert}} \in \underset{\theta \in \mathbb{R}^P}{\arg\min} \, L(\theta, \Psi + \Delta) \tag{8b}$$

$$\theta_{\text{org}} \in \underset{\theta \in \mathbb{R}^P}{\arg\min} \, L(\theta, \Psi) \tag{8c}$$

$$\|\delta_i\|_p \leq \varepsilon, \quad i = 1, \ldots, N \tag{8d}$$

$$\sum_{i=1}^n \mathbf{1}_{\|\delta_i\| \neq 0} \leq \alpha \cdot n. \tag{8e}$$

The DOPE objective in (8a) increases the estimated value from the original value, thereby increasing the error. Alternatively, if the attacker wants to decrease the estimated value of the given policy, they may do so by simply changing the sign of the objective. The constraint (8b) estimates the optimal parameter $\theta_{\text{pert}}$ from $D$ after perturbing $\Psi$ to $\Psi + \Delta$. The constraint (8d) ensures that the perturbation added to each sample $\delta_i$, i.e. $i^{\text{th}}$ row of $\Delta$, is limited to the user-defined budget $\varepsilon$ in $\ell_p$ norm. This prevents the attack framework from generating adversarial transitions that can be easily detected as anomalous. Further, the constraint (8e) limits the number of transitions that the attacker can perturb. Finally, note that $\theta_{\text{org}}$ is only computed once with the original features $\Psi$ and $\rho(\theta_{\text{org}}, \Psi)$ is a constant that can be ignored while solving the optimization problem.

## 3.1 ATTACKING OPE METHODS USING DOPE

We are now ready to formally define the four components of the DOPE framework. Table 1 summarizes the choice of these components for each OPE method we attack.
(a) *Features*: Let $\psi(s, a, r) \in \mathbb{R}^Q$ be an arbitrary component of the transition tuple $\langle s, a, r \rangle$ in $D$ that is perturbed by the attacker. For example, $\psi(s, a, r)$ could either be the state features $\Phi$ or the reward vector $r$. We will use $\Psi \in \mathbb{R}^{n \times Q}$ to represent the sample matrix of $\psi(s, a, r)$ constructed from the $n$ samples in $D$.
(b) *Parameters*: The parameters $\theta(\Psi) \in \mathbb{R}^P$ are the parameters of interest for a given OPE method, written as a function of $\Psi$ to clarify that these are estimated from samples in $D$. In BRM, $\theta$ represents the parameters of the Q-value function $q_\eta(s, a)$, whereas in IS methods, $\theta$ represents the parameters of the estimated behavior policy $\pi_b^{\theta_b}(a|s)$.
(c) *Loss function*: The loss function $L(\theta, \Psi)$ with $L \colon \mathbb{R}^P \times \mathbb{R}^{n \times Q} \to \mathbb{R}$ is the empirical loss optimized by the OPE method to derive the optimal parameter $\theta(\Psi) \in \arg\min_{\theta' \in \mathbb{R}^P} L(\theta', \Psi)$ from the data. As an example, $L$ in BRM and DR is the MSBR error, whereas in IS methods, $L$ is the CEL loss optimized to estimate the behavior policy.
(d) *Value estimation function*: Finally, the value estimation

function $\rho(\theta(\Psi), \Psi)$ with $\rho \colon \mathbb{R}^P \times \mathbb{R}^{n \times Q} \to \mathbb{R}$ is the function used by the OPE method to compute the mean value of $\pi$ at the initial states. For example, in BRM, $\rho$ represents $v_{\text{BRM}}$. We will use the shorthand $\rho(\Psi) := \rho(\theta(\Psi), \Psi)$.

The loss function $L(\theta, \Psi)$ must be twice continuously differentiable and linearly separable with respect to the transitions in $D$. We provide some examples of such loss functions such as MSBR and CEL and show that they are twice continuously differentiable in Section 4. Further, the value estimation function $\rho(\theta, \Psi)$ also needs to be continuously differentiable with respect to $\theta$ and $\psi$. These assumptions, as Section 4 shows, are important for the influence functions to be well-defined [Koh and Liang, 2017].

## 4 OPTIMIZATION

In this section, we discuss the challenges of optimizing the DOPE problem in (8) and propose an approximate scheme for finding the optimal adversarial perturbations.

There are two major challenges in solving the optimization problem in Equation (8). First, the constraint (8e) is non-differentiable and requires the attacker to select a set of at most $\alpha n$ transitions, such that perturbing these transitions results in maximum change in the value of the policy in the desired direction. It is important to realize that finding this set requires perturbing all possible subsets of data $\Psi$ whose size is at most $\alpha n$ and re-estimating the optimal parameter $\theta$ for each perturbation. The number of such subsets is larger than $\binom{n}{\alpha n}$. Thus, finding the optimal set is computationally infeasible. Second, observe that (8) is a bilevel optimization problem where the inner-level problem (8b) is non-linear in the case of OPE methods which makes it generally NP-Hard to solve [Wiesemann et al., 2013].

We address these two challenges by deriving an approximation to the bilevel optimization problem (8) using the Taylor expansion of Equation (8a). We show that the resulting problem is simpler to optimize and has a closed-form solution. In Section 5, we empirically demonstrate the effectiveness of our approximate solution on several domains.

**Approximation** We define the influence score of the $i^{\text{th}}$ data point as $I_{\Psi_i} = \nabla_{\Psi_i} \rho(\Psi)$ as the rate of change in the value estimate $\rho(\Psi)$ with respect to the data point $\Psi_i \equiv \psi(s_i, a_i, r_i)$. Then, using the first-order Taylor expansion of $\rho(\Psi + \Delta)$, we can approximate the net error in the value-function estimate $\rho(\Psi + \Delta) - \rho(\Psi)$ as the weighted sum of the influence scores of individual data points,

$$\rho(\Psi + \Delta) - \rho(\Psi) \approx \sum_{i=1}^n (\nabla_{\Psi_i} \rho(\Psi))^\top \delta_i. \tag{9}$$

| Method | Parameters $\theta$ | Features $\Psi$ | Function $\rho(\Psi)$ | Loss $L(\theta, \Psi)$ |
|---|---|---|---|---|
| BRM (Farahmand et al. [2008]), Eq. (3) | $\eta$ in $q_\eta$ | $\Phi$ or $r$ | $v_{\text{BRM}}$ | MSBR |
| WIS (Rubinstein [1981]), Eq. (4) | $\theta_b$ in $\pi_b^{\theta_b}$ | $\Phi$ or $r$ | $v_{\text{WIS}}$ | CEL |
| PDIS (Precup [2000]), Eq. (5) | $\theta_b$ | $\Phi$ or $r$ | $v_{\text{PDIS}}$ | CEL |
| CPDIS (Thomas [2015]), Eq. (6) | $\theta_b$ | $\Phi$ or $r$ | $v_{\text{CPDIS}}$ | CEL |
| WDR/DR (Jiang and Li [2016]), Eq. (7) | $\theta_b, \eta$ | $\Phi$ or $r$ | $v_{\text{WDR}}$ or $v_{\text{DR}}$ | CEL + MSBR or MSBR |

**Table 1:** Settings for the four components of the DOPE attack for five different OPE methods.

Using Eq. (9) reduces the optimization in (8) to

$$\max_{s \in \{0,1\}^n} \max_{\{\delta_i\}_{i=1}^N} \left\{ \sum_{i=1}^n s_i I_{\Psi_i}^\top \delta_i \mid \|\delta_i\|_p \leq \varepsilon, \forall i \right\},$$

$$\text{subject to} \sum_{i=1}^n s_i = \alpha \cdot n . \tag{10}$$

Here, $s \in \{0,1\}^N$ is a vector of binary indicators such that $s_i = 1$ indicates that the $i^{\text{th}}$ transition is amongst the $\alpha n$ transitions selected to be perturbed. We can now compute an approximately optimal set of perturbations in polynomial time as shown in Theorem 4.1 for norms $p = 1, 2, \infty$.

**Theorem 4.1.** *Let $(s^*, \Delta^*)$ be an optimal solution to the optimization problem in* (10) *and define the* approximate influential set *as $S_\alpha^* = \{i : s_i^* = 1, \forall i = 1, \ldots, n\}$. Then,*

1. *$S_\alpha^*$ can be constructed by choosing the set of $\alpha n$ transitions with the largest q-norm of their influence scores $I_{\Psi_i}$. Here, q-norm is the dual of p-norm used in* (10), *i.e. $1/p + 1/q = 1$.*
2. *For all $i \in [1, \ldots n]$, the optimal $\delta_i^*$ for $p = 1, 2, \infty$ can be computed in closed-form as*

   *If $p = \infty$, then $\delta_i^* = \varepsilon \cdot \text{sign}(I_{\Psi_i})$*

   *If $p = 2$, then $\delta_i^* = \varepsilon \cdot \dfrac{I_{\Psi_i}}{\|I_{\Psi_i}\|_2}$.*

   *If $p = 1$, then $\forall j \in [1, Q]$,*

   $$\delta_{i,j}^* = \begin{cases} \varepsilon \cdot \text{sign}(I_{\Psi_i}(j)) & \text{if } j \in \arg\max_{m \in [1,Q]} I_{\Psi_i}(m) \\ 0 & \text{otherwise} \end{cases}$$

*Remark* 4.2 (Relation to optimal solution). Solving the approximate problem (10) gives us a lower bound to the optimal solution of the original problem (8). Suppose $\Delta^*$ is the optimal solution for (10) that we get from Theorem 4.1 while $\Delta^{**}$ is the (intractable) optimal solution for (8). Then, the maximum error in the value function is at least as much as what we get,

$$\rho(\Psi + \Delta^{**}) - \rho(\Psi) = \max_{\Delta \in \mathbb{R}^{n \times Q}} \rho(\Psi + \Delta) - \rho(\Psi)$$

$$\geq \rho(\Psi + \Delta^*) - \rho(\Psi) .$$

**Influence scores** Finally, it remains to discuss how to compute the influence scores of each transition in $D$: $I_{\Psi_i} =$ $\nabla_{\Psi_i} \rho(\Psi)$. Recall that $\rho(\Psi)$ is not only a function of $\Psi$ but also $\theta(\Psi)$ which is also a function of $\Psi_i$. Hence, using the chain rule, we get for each $i \in [1 \ldots n]$ that

$$I_{\Psi_i} \approx \frac{\partial \rho(\theta, \Psi)}{\partial \Psi_i}\bigg|_{\theta_{\text{org}}(\Psi)} + \frac{\partial \rho(\theta, \Psi)}{\partial \theta}\bigg|_{\theta_{\text{org}}(\Psi)} \frac{\partial \theta(\Psi)}{\partial \Psi_i}. \tag{11}$$

---

**Algorithm 1:** OPE Attack Algorithm

**Input:** Features $\Psi$, attack budget $\varepsilon$, % of corrupt transitions $\alpha$, norm-type $p$

Compute $\theta_{\text{org}} \leftarrow \arg\min_{\theta \in \mathbb{R}^P} L(\theta, \Psi)$ ;

Compute $\|I_{\Psi_i}\|_q$ for all $i = 1, \ldots, n$ using (11) ;

$S_\alpha^* \leftarrow \alpha \cdot n$ indices $i$ with largest $\|I_{\Psi_i}\|_q$ ;

**for** $k \in S_\alpha^*$ **do**

  Let $\delta_k^* \in \arg\max_{\delta \in \mathbb{R}^Q} \{I_{\Psi_k}^\top \delta \mid \|\delta\|_p \leq \varepsilon\}$ using Item 2
  in Theorem 4.1;

**end**

Use line search to find the largest step-size $\beta \in [0, 1]$
  s. t. the value estimate increases:
  $\rho(\theta, \Psi + \beta \cdot \delta^*) - \rho(\theta, \Psi) > 0$;

**return** $\Psi = \Psi + \beta \cdot \delta^*$ ;

---

The computation of the partial derivative $\partial \theta(\Psi)/\partial \Psi_i$ is not straightforward. However, we can approximately compute it as $\partial \theta(\Psi)/\partial \Psi_i = H_{\theta_{\text{org}}(\Psi)}^{-1} \partial^2 L(\theta, \Psi_i)/\partial \theta \partial \Psi_i \big|_{\theta_{\text{org}}(\Psi)}$ where $H_{\theta_{\text{org}}(\Psi)} = \partial^2 L(\theta, \Psi)/\partial \theta^2 \big|_{\theta_{\text{org}}(\Psi)}$ [Koh and Liang, 2017, Section 2.2]. See Section 1 for more details.

To compute $I_{\Psi_i}$ in (11), we require that $L(\theta, \Psi)$ is twice continuously differentiable and linearly separable with respect to the transitions in $D$, and $\rho(\theta, \Psi)$ is continuously differentiable with respect to $\theta$ and $\psi$. Although these conditions may seem restrictive, they hold true for the OPE methods we have studied.

The derivatives in (11) can be easily computed using automatic-differentiation software like PyTorch [Paszke et al., 2019]. Computing the influence score $I_{\Psi_i}$ can be very expensive due to the Hessian-inverse term $H_{\theta_{org}(\Psi)}^{-1}$ which requires $\mathcal{O}(P^3)$ operations to compute. Fortunately, as shown in [Koh and Liang, 2017], we can avoid the computation of the Hessian-inverse term while computing $I_{\Psi_i}$ by instead first approximately computing the Hessian-inverse

vector product

$$c_{\text{prod}} = H_{\theta_{\text{org}}(\Psi)}^{-1} \left. \frac{\partial \rho(\theta, \Psi)}{\partial \theta} \right|_{\theta_{\text{org}}(\Psi)}$$

in $\mathcal{O}(nP)$ time using the Pearlmutter's method [Pearl-mutter, 1994] and first-order Taylor approximation of the Hessian-inverse matrix, and then applying the Pearlmutter's method again to compute the Hessian-vector product $c_{\text{prod}} \cdot \partial^2 L(\theta, \Psi_i) / \partial\theta\partial\Psi_i \big|_{\theta_{\text{org}}(\Psi)}$ in $\mathcal{O}(P)$ time.

**Algorithm outline** We outline how to approximately solve the DOPE optimization (8) in Algorithm 1, which consists of two main steps. In the first step, we compute an approximation of the optimal set of transitions to perturb $S_\alpha^*$ by choosing $\alpha n$ points in $\Psi$ with the largest $q$-norm of their influence scores $\|I_{\psi.}\|_q$. In the second step, we compute $\Delta$ for all points in $S_\alpha^*$ using Theorem 4.1 and use line search to find the optimal step size that guarantees an increase in the error of the value estimate. The second step may be repeated until no further perturbation to data points in $S_\alpha^*$ results in an increase in the error in the value estimate.

The main computational bottleneck is in computing the influence score for each data point. In some cases, this cost can be reduced. We derive closed-form expressions for the influence score in the case of the linear BRM method under two settings a) when the adversary perturbs only the state features or b) only the reward features.

**Proposition 4.3.** *If the attacker only perturbs the reward vector $r$ constructed from batch of transition tuples $D$. Then, the influence score of the $i^{th}$ data point $I_{r_i, \theta, \Psi}$ for the BRM method can be computed as*

$$I_{r,\theta,\Psi} = 4p_0^T \Phi_0 \left( (\Phi - \gamma \cdot \Phi_p)^2 \right)^{-1} (\Phi - \gamma \cdot \Phi_p) , \quad (12)$$

*where $\Phi_0$ is a sample matrix of initial state features constructed from $D_0$.*

**Proposition 4.4.** *If the attacker only perturbs the state feature matrix $\Phi$. Then, the influence score of the $i^{th}$ data point $I_{\phi(s_i, a_i), \theta, \Psi}$ for the BRM method can be computed as*

$$\begin{aligned} I_{\Psi,\theta,\Psi} = {} & 4 \cdot p_0^T \cdot \Phi_0 \cdot \left( (\Phi - \gamma \cdot \Phi_p)^2 \right)^{-1} \cdot \\ & \left( 2 \cdot w \bigotimes \Phi - 2 \cdot \gamma \cdot w \bigotimes \Phi_p \right. \\ & \left. + 2 \cdot \mathbf{I} \bigotimes (\Phi \cdot w - \gamma \cdot \Phi_p \cdot w - r) \right), \end{aligned} \quad (13)$$

*where $\bigotimes$ denotes the Kronecker product between matrices.*

Proposition 4.3 and 4.4 follow from the chain rule and basic properties of the gradient operator for matrices.

# 5 EXPERIMENTS

In this section, we investigate the strengths and weaknesses of the DOPE attack. First, we evaluate the effectiveness of the DOPE attack on OPE methods for different values of attack budget and identify which methods are most vulnerable to adversarial contamination. Second, we compare the performance of DOPE with three custom baselines: Random DOPE, FSGM-based Attack, and Random Attack.

## 5.1 DOMAINS AND EXPERIMENTAL SETUP

We first describe the five domains used in the experiments. *Cancer:* This domain [Gottesman et al., 2020] models the growth of tumors in cancer patients. It consists of 4-dimensional states which represent the growth dynamics of the tumor in the patient, and two actions that indicate if a given patient is to be administered chemotherapy or not at a given time step.
*HIV:* The HIV domain [Ernst et al., 2006b] has 6-dimensional states representing the state of the patient, and four actions that represent four different types of treatments.
*Mountain Car:* In the Mountain Car [Brockman et al., 2016] domain, the task is to drive a car positioned between two mountains to the top of the mountain on the right in the shortest time possible. The 2-dimensional state represents the car's current position and the current time-step, and the three actions represent: drive forward, drive backward, and do not move.
*Cartpole:* The Cartpole domain [Brockman et al., 2016] models a simple control problem where the goal is to apply +1/-1 force to keep a pole attached to a moving cart from falling. The 2-dimensional state represents the cartpole dynamics, and the two actions represent the force applied to the pole.
*Continuous Gridworld:* The Continuous Gridworld is a custom domain that consists of a 2-dimensional state space that represents the coordinates of the agent, and two actions $a_0, a_1$ that determine the direction and step size of the agent. The agent begins at coordinate $(1, 1)$ and moves towards coordinates $(50, 50)$ to maximize its rewards.

**Implementation details** For each domain, we apply Deep Q-learning (DQN) to a randomly initialized neural network policy and obtain partially optimized deterministic policies. We fix the deterministic policy obtained for each domain as the evaluation policy and use an $\epsilon$-greedy version of the evaluation policy as our behavior policy [Gottesman et al., 2020]. We set $\epsilon = 0.1$ for the HIV domain and $\epsilon = 0.05$ for other domains. We use the behavior policy to generate five datasets, each containing $N$ trajectories of length $T$ (see Section 3 for the values of $N$ and $T$) and use it to evaluate the value of the evaluation policy. Our code is made available in the supplementary materials.

For any given OPE method that learns the Q-value function of the evaluation policy from data, we use linear function approximators to represent these Q-value functions and optimize the squared Bellman residual with $L_2$ regularization

to learn it. Note that we consider linear function approximations in line with the precedent set by other recent works in the off-policy evaluation literature [Gottesman et al., 2020, Jin et al., 2020]. Linear function approximations are commonly employed in the off-policy evaluation literature due to their simplicity, low computational complexity, and convergence guarantees [Gottesman et al., 2020]. While our framework is general enough to accommodate any differentiable function approximations including deep learning models, computing the influence functions for non-linear function approximations is computationally expensive, and the time complexity grows as the square of the number of model parameters $\theta$. Hence, we resort to linear function approximators. Note, however, that we try to offset the limitations in the expressive power of linear function approximations by leveraging complex state representations obtained from the second last layer of a trained deep Q-network as input features to the linear function approximations in our experimentation (See Section 3).

For OPE methods that require learning behavior policy from the data, we train a multinomial logistic regression model to predict the behavior policy's action probabilities using maximum likelihood estimation. Following standard practice in RL, we clip the behavior probabilities to $0.01$ to avoid importance sampling weights from exploding. Note that although clipping the behavior probabilities prohibits the attacker from making individual behavior policy action probabilities too small, an attacker can still leverage the fact that the importance sampling weights are a function of the product of behavior policy action probabilities, and thus, the importance weights can be made very large by simply making the behavior policy action probabilities of as many points as possible to close to the clipping threshold.

In all our experiments, we perturb only state features. Finally, the values of the hyperparameters used in our experiments are discussed in Section 3.

We evaluate the effectiveness of our attack model by computing the percentage error in the value function estimate relative to the initial value estimate. We report the $95\%$ bootstrap confidence intervals of the interquartile mean (IQM) of percentage error using our results from the 10 runs (10 datasets) since the IQM confidence intervals are found to be more reliable in practice [Agarwal et al., 2021]. In this setting, a large percentage error indicates that the OPE method is less robust to adversarial contamination.

## 5.2 EFFECTIVENESS OF DOPE ATTACK

Here we evaluate the effectiveness of the DOPE attack on five OPE methods for a range of attack budgets. In our first experiment, we fix the percentage of corrupt data points $\alpha = 0.05$ and vary the budget $\varepsilon$ as frac $\cdot\ \sigma$, where frac varies from $0.0$ to $0.51$ in step-sizes of $0.05$ and

$\sigma^2 = \frac{2}{N \cdot (N-1)} \sum_{i=1}^{N} \sum_{j=i+1}^{N} \|\xi(s_i) - \xi(s_j)\|_p^2$ is the standard deviation of all pairwise distances between the state-features in the dataset. Figure 1 compare the percentage error in the value estimate of the OPE methods in all domains. Our results show that even when corrupting only $5\%$ of the data points, the attacker need not perturb the state features significantly to achieve large errors in the value estimate. In fact, with a perturbation budget as small as $\varepsilon = 0.5\sigma$, DOPE can result in a substantial error in the policy's value in HIV, Cancer, Mountain Car, and Continuous Gridworld domains. Further, a larger attacker's budget means the DOPE model has more leeway on the perturbations that it can add to the dataset, and hence, we observe larger errors for larger budget values. Note that the percentage errors of CPDIS and WIS in Figures 1b and 1c are too small to be visible in the plots.

In the second experiment, we vary the percentage of corrupt data points between $0.0$ and $0.10$ with a step size of $0.02$ for all the domains (Figure 2). We fix the perturbation budget $\varepsilon$ to $1.0\sigma$. Our experimental results in Figures 1 to 2 demonstrate that corrupting only $0.05\%$ of the data points using DOPE is sufficient to observe a significant error in the value estimate of a given policy. It is important to realize that the attacker's budget $\varepsilon$ is local to each data point and is not impacted by the number of points perturbed. Hence, we see that a larger percentage of corrupt data points yields a larger percentage error in the value estimates. Note that the percentage errors of CPDIS and WIS in Figures 2b and 2c are too small relative to BRM and WDR and therefore are not clearly visible in the plots.

Finally, we summarize the impact of DOPE attack ($\varepsilon = 0.5\sigma$ and $\alpha = 1.0$, $p = 1$) on all OPE methods and domains in Table 2. It can be seen that the DOPE attack has a very high impact on BRM, PDIS, and WDR methods and an almost negligible impact on CPDIS and WIS methods. We hypothesize that CPDIS and WIS methods may be more robust because the weight normalization that they employ potentially minimizes the importance of any individual data point, especially when the rewards are uniformly distributed throughout the trajectory. On the other hand, the weights in PDIS are not normalized, and therefore, it appears that in Cartpole and HIV domains, the DOPE attack model is able to significantly impact the importance sampling weights and result in significant errors in the value estimates. In WDR, the attacker can introduce errors through both, the Q-value function learned from the data as well as the importance sampling weights, and therefore, we observe significant errors in the value estimates of WDR method in HIV and Gridworld domains.

## 5.3 COMPARISON WITH BASELINES

Here we compare the DOPE attack to three custom baselines: Random Attack and Random DOPE Attack (ablation

| Domain | BRM | | WIS | | PDIS | | CPDIS | | WDR | |
|---|---|---|---|---|---|---|---|---|---|---|
| | *lb* | *ub* | *lb* | *ub* | *lb* | *ub* | *lb* | *ub* | *lb* | *ub* |
| Cancer | 0.85 | 0.97 | 0.69 | 0.69 | **8.95** | **10.69** | 0.48 | 0.58 | 3.36 | 3.72 |
| HIV | **343.35** | **440.92** | 0.0 | 0.1 | 1.4 | 2.42 | 0.09 | 0.24 | **139.71** | **893.31** |
| Gridworld | **94.76** | **98.35** | 0.0 | 0.0 | **97.15** | **98.25** | 0.0 | 0.0 | **25.5** | **27.31** |
| Cartpole | 0.0 | 0.0 | 0.02 | 0.05 | **4.46e9** | **4.08e10** | 0.0 | 0.0 | 0.0 | 0.0 |
| MountainCar | 0.05 | 0.07 | **100.0** | **100.0** | 98.37 | 99.62 | 47.38 | 98.68 | 0.02 | 0.03 |

**Table 2:** Summary of the errors achieved by data poisoning across domains and OPE algorithms at $\varepsilon = 0.5\sigma$ and $\alpha = 1.0$ and $p = 1$. Here *lb* and *ub* denote the lower limit and upper limit of 95% bootstrap confidence intervals of interquartile mean of percentage error in the value estimates, over 10 runs. We observe that the attack is successful on most of the methods across domains. CPDIS and WIS are usually the most resilient OPE methods.

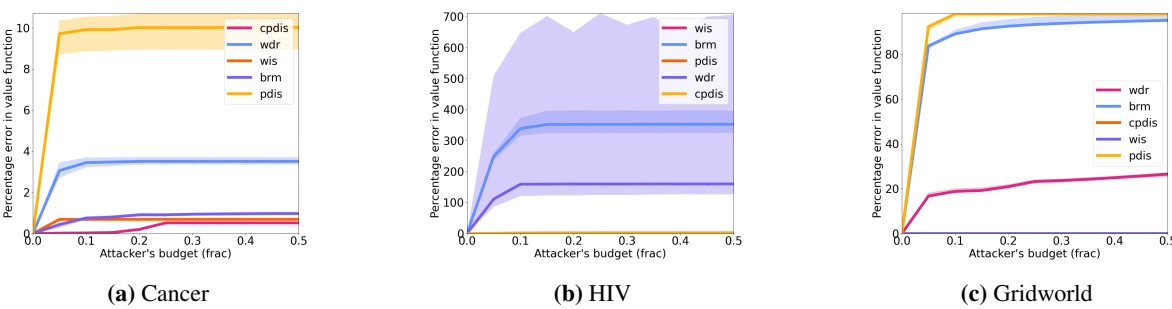

| (a) Cancer | (b) HIV | (c) Gridworld |
|---|---|---|

**Figure 1:** Figures 1a to 1c compares the effect of DOPE attack on BRM, WIS, PDIS, CPDIS and WDR methods in the Cancer, HIV and Continuous Gridworld domains (left to right) for different values of attacker's budget $\varepsilon = \text{frac} \cdot \sigma$ and $p = 1$ ($\ell_1$ norm). Larger the value of frac, the larger are the perturbations added by the DOPE attack, and accordingly we observe larger errors in the value estimates.

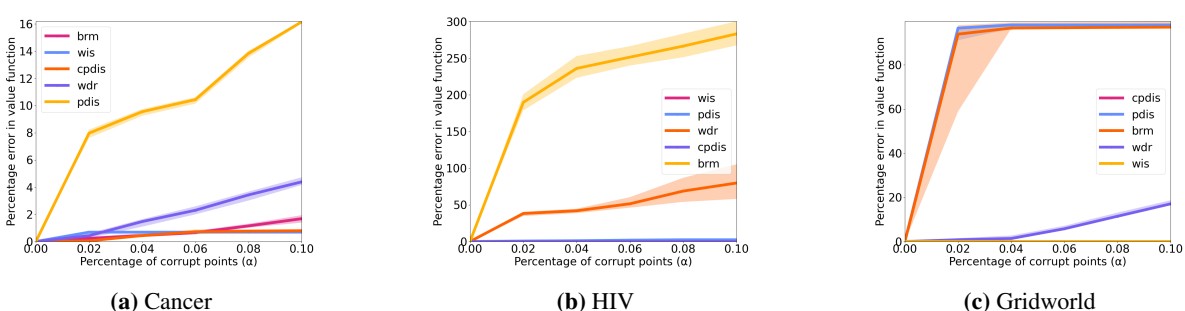

| (a) Cancer | (b) HIV | (c) Gridworld |
|---|---|---|

**Figure 2:** Figures 2a to 2c compares the effect of DOPE attack on BRM, WIS, PDIS, CPDIS, and WDR methods in Cancer, HIV, and Continuous Gridworld domains (left to right) for different percentages of corruption $\alpha$ at $\varepsilon = 1.0\sigma$ and $p = 1$ ($l_1$ norm). The larger the value of $\alpha$, the larger the number of points perturbed by the DOPE attack, and accordingly, we observe larger errors in the value estimates.

of DOPE Attack) and FSGM-based Attack. In Random Attack, we choose $\alpha n$ random points to perturb and sample perturbations for these points from a uniform $l_1$ norm ball with a radius equal to the perturbation budget $\varepsilon$. For more details on the sampling algorithm, see Algorithm 4.1 in [Calafiore et al., 1998]. In Random DOPE Attack, we select points randomly and update them using Theorem 4.1. The purpose of using this ablation is to investigate the benefit of selecting data points to perturb based on their influence scores as suggested in Theorem 4.1. The third baseline is an FGSM-based OPE attack which is a variant of the Fast Gradient Sign Method (FGSM) [Goodfellow et al., 2015], a popular

test-time attack designed to elicit misclassification errors from supervised learning models. Note that FGSM has never been used to attack OPE methods in prior literature, and we are the first to introduce and leverage a variant of it as a baseline in this context. Our FGSM-based OPE attack baseline modifies the transition tuples (features) $\psi(s, a, r)$ to maximize the (supervised learning) loss $(L(\theta))$ optimized by the OPE method, thus resulting in sub-optimal estimates of $\theta$. Note that the FGSM-based OPE attack baseline does not directly maximize the error in the value function estimates, unlike our proposed framework. Given these baselines, we fix the value of $\alpha$ to 0.05 and vary the budget $\varepsilon$ from 0.0 to

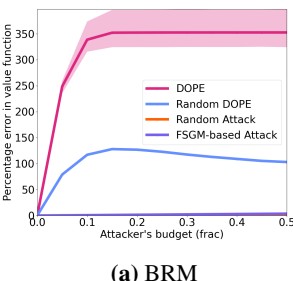
**(a)** BRM

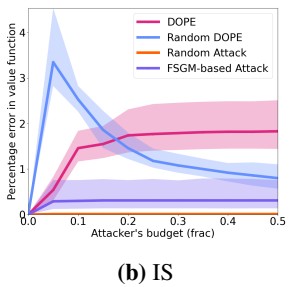
**(b)** IS

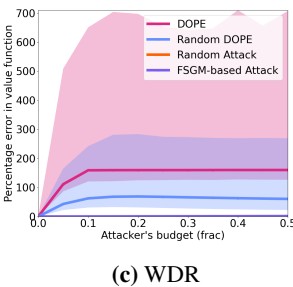
**(c)** WDR

**Figure 3:** Figures 3a to 3c compare the effects of Random attack, Random DOPE attack (an ablated version of DOPE), FSGM-based Attack and DOPE attack on the error in the value function estimates of BRM, IS, and DR methods (left to right) in HIV domain. The percentage error in the Random attack and FSGM-based attack is small relative to the percentage error due to DOPE and Random DOPE attack, and hence their curves lies close to the x-axis. DOPE attack outperforms both the Random DOPE and Random attacks at nearly all values of the attacker's budget.

0.25 with step size 0.04.

For each dataset and each value of the budget $\varepsilon$, we average the percentage change in the value estimate for Random DOPE attack and Random attack over 50 trials. Results with the Gridworld domain are shown in Figure 3. See Figures 3 and 5 to 7 in Section 3 for results on other datasets.

The experimental results in Figures 3 and 5 to 7 demonstrate that in contrast to the DOPE attack, the Random attack and FSGM-based attack fail to introduce any significant error in the value-function estimate and, therefore, cannot be used as an alternative to the DOPE attack model. Further, it can be seen that when the points to perturb are randomly selected (Random DOPE), it is likely to result in a smaller adversarial impact than when influential data points are chosen for perturbations (DOPE). These results are not surprising as we would expect the value function to be highly dependent on the influential data points. In some domains like Cancer and HIV, there is very little difference between the performance of DOPE and Random DOPE attacks. We hypothesize that this is due to all data points having similar influence scores.

# 6 CONCLUSION

We proposed a novel data poisoning framework to analyze the sensitivity of OPE methods to adversarial contamination at train time. We formulated the data poisoning problem as a bilevel optimization problem and proposed a computationally tractable solution that leverages the notion of influence functions from robust statistics literature. Using the proposed framework, we analyzed the sensitivity of five popular OPE methods on multiple datasets from medical and control domains. Our experimental results on various medical and control domains demonstrated that existing OPE methods are highly vulnerable to adversarial contamination thus highlighting the need for developing OPE methods that are statistically robust to train-time data poisoning attacks.

## Acknowledgements

The authors would like to thank the anonymous reviewers for their helpful feedback and all the funding agencies listed below for supporting this work. This work is supported in part by the NSF awards #IIS-2008461 and #IIS-2040989, and research awards from Amazon, Harvard Data Science Institute, Bayer, and Google. HL would like to thank Sujatha and Mohan Lakkaraju for their continued support and encouragement. The views expressed here are those of the authors and do not reflect the official policy or position of the funding agencies.

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
