# OpenReview forum: "Data Poisoning Attacks on Off-Policy Policy Evaluation Methods"
_auai.org/UAI/2022/Conference — UAI 2022 Oral_

### Official Review · Reviewer_QijX · 2022-04-10

**Q2(1) Originality/Novelty:** 2
**Q2(2) Significance/Impact:** 3
**Q2(3) Correctness/Technical Quality:** 3
**Q2(6) Clarity Of Writing:** 4
**Q6 Overall Score:** 6
**Q8 Confidence In Your Score:** 3

**Q1 Summary And Contributions:**

The paper studies the problem of modifying the training data (i.e., poisoning attack) of an off-line RL method such that the estimated value functions deviate from the true functions in a point-wise fashion.  The problem is formulated as a bi-level optimization (BLO), a common strategy to model poisoning attacks. As bi-level optimization problems are challenging, the paper introduces a novel heuristic for this involving first-order Taylor and influence functions.

**Q2 Assessment Of The Paper:**

More detailed information regarding each of these aspects is given below:

**Q2(4) Quality Of Experiments (Optional):**

3: Good: The experimental evaluation is adequate, and the results convincingly support the main claims.

**Q2(5) Reproducibility:**

3: Good: Key resources (e.g., proofs, code, data) are available and key details (e.g., proofs, experimental setup) are sufficiently well-described for competent researchers to confidently reproduce the main results.

**Q3 Main Strengths:**

1. The paper is well-motivated. Due to the scarcity of data, it may be inevitable that we have to utilize off-line RL methods to estimate the value functions. The adversarial/security issues thus become critical, especially in life-critical settings (e.g., vehicle control).

2. The formulation of the poisoning attack as a BLO is natural. Although the BLO is hard to solve, a tractable version is provided; a heuristic algorithm is designed to solve the tractable version.

3.The experiments are good and show the efficacy of the attacking.

4.The paper is well-written and easy to follow.

**Q4 Main Weakness:**

The novelty is somewhat limited from the modeling side. Given the data D of multiple trajectories, the off-line RL methods are essentially doing standard supervised learning. It would be direct to come up with the bi-level model. That said, the paper has contributions on transferring the hard-to-solve bi-level model to a tractable optimization problem by linearizing the objective function with Taylor expansion and influence function scores.

**Q5 Detailed Comments To The Authors:**

See strengths/weaknesses above.  Here are several additional comments:

1. Some notation is misleading; for example, a policy would be \pi(a | s), instead of \pi(s, a).

2. It seems that the trajectories in the dataset D are IID. What if this assumption is not true? For example, suppose you have a single (very) long trajectory and you somehow chop it into sub-trajectories with equal lengths; this gives you a new data D’ with non-IID data; how does this affect your poisoning strategy?  I think the non-IID case would not be rare in practice, especially in the settings where you can’t just simulate multiple trajectories.


**Q7 Justification For Your Score:**

See comments above.

**Q9 Complying With Reviewing Instructions:**

1: Yes.

---

### Official Review · Reviewer_eLuh · 2022-04-11

**Q2(1) Originality/Novelty:** 3
**Q2(2) Significance/Impact:** 3
**Q2(3) Correctness/Technical Quality:** 4
**Q2(6) Clarity Of Writing:** 4
**Q6 Overall Score:** 8
**Q8 Confidence In Your Score:** 4

**Q1 Summary And Contributions:**

The paper describes an approach called DOPE for data poisoning in the context of policy evaluation.  DOPE is based on bi-level optimization and is applicable to several off-policy evaluation methods including BRM, WIS, PDIS, CPDIS, DR and WDR. While some of them are more robust than others, the paper shows the need to design more robust methods.

**Q2 Assessment Of The Paper:**

More detailed information regarding each of these aspects is given below:

**Q2(4) Quality Of Experiments (Optional):**

4: Excellent: The experimental evaluation is comprehensive and the results are compelling.

**Q2(5) Reproducibility:**

3: Good: Key resources (e.g., proofs, code, data) are available and key details (e.g., proofs, experimental setup) are sufficiently well-described for competent researchers to confidently reproduce the main results.

**Q3 Main Strengths:**

The paper introduces an important problem, namely data poisoning of off-policy evaluation.
The bi-level optimization approach is principled, well-justified and is applicable to a number of methods for off-policy policy evaluation.
The empirical results are thoroughly done. DOPE has been illustrated for 5 different methods of off-policy evaluation in 5 different domains including some real world domains. The effect of hyperparameters has been well documented.
The paper is well organized and well written.

**Q4 Main Weakness:**

The baseline method (random) is too weak and does not show how good DOPE really is. Are there other state of the art methods to compare against?

In the experiments the behavior policy is close to the evaluation policy policy. It would be interesting to study larger values of epsilon to see how the results change.





**Q5 Detailed Comments To The Authors:**

Page 3 has some typos. "behavior policy is parameterized by \theta" . You later use \theta_b.

Equation 4 has L instead of T.

Last line of the first column.    v_\eta (s_t^i) = \sum ....
The rhs of this equation has s in it and the lhs has s_t^i. I believe they should be the same.
Second column second sentence of page 3. "show that THERE are no clear winners"

I like Table 1. Thank you.


**Q7 Justification For Your Score:**

The paper introduces the problem of data poisoning in the context of policy evaluation, describes a principled approach for it and evaluates it thoroughly in multiple domains. The paper is well written.

**Q9 Complying With Reviewing Instructions:**

0: No.

---

### Official Review · Reviewer_3xVH · 2022-04-12

**Q2(1) Originality/Novelty:** 3
**Q2(2) Significance/Impact:** 3
**Q2(3) Correctness/Technical Quality:** 3
**Q2(6) Clarity Of Writing:** 4
**Q6 Overall Score:** 8
**Q8 Confidence In Your Score:** 3

**Q1 Summary And Contributions:**

This article proposes a framework for perturbing training data in off-policy evaluation (OPE) techniques and evaluate the robustness of such perturbation on the policy value estimates. They propose the identification of specific perturbation as an optimization problem, relax the optimization problem for computational feasibility, and use influence functions to identify the required perturbations. They evaluate the effect of their perturbation on various well known OPE methods.

**Q2 Assessment Of The Paper:**

More detailed information regarding each of these aspects is given below:

**Q2(4) Quality Of Experiments (Optional):**

3: Good: The experimental evaluation is adequate, and the results convincingly support the main claims.

**Q2(5) Reproducibility:**

2: Fair: Key resources (e.g., proofs, code, data) are unavailable but key details (e.g., proof sketches, experimental setup) are sufficiently well-described for an expert to confidently reproduce the main results.

**Q3 Main Strengths:**

1. Well-written article with a clear flow of ideas.
2. The optimization problem and corresponding relaxation to improve computational complexity.
3. Generalizable framework for data poisoning and careful attention to fine tune the framework for various OPE techniques.


**Q4 Main Weakness:**

1. The main issue with this framework is the motivation for data poisoning. How likely is to have access to various parameters of the OPE technique and being able to perform the data poisoning?
2. Evaluation of the distinction between \Psi + \Delta and \Psi by anomalous detection methods.
3. Empirical comparison to more involved methods than Random and Random DOPE


**Q5 Detailed Comments To The Authors:**

1. The authors need to motivate the methodology by providing more details on where these perturbation techniques are practical in real-world settings. We probably also need to see if we can identify the perturbation introduced by using anomaly detection techniques. Though the perturbations can change a policy's value function, if they can be easily identified, decision-makers can run these techniques to check if there are any anomalies. More interestingly, we can try and introduce perturbations that make it harder to be detected and change the value functions the most. One way to do this is to add additional constraints to the optimization problem.
2. The empirical evaluation can be improved by introducing a more sophisticated strawman. This is primarily motivated by the Random DOPE is already performing very well, increasing (or indistinguishable to the DOPE) in lower frac values (in Figures 3, 6, and 7). This comparison will further enhance the need for using the complex optimization framework proposed by the authors.
3. Minor typo: The sentence after equation 10 should be n instead of N, that is, “Here, $s \in \{0, 1\}^n$ is a vector….”
4. Minor: Is the BRM visualization missing in Figures 1 and 2? It is not even present in the legend.


**Q7 Justification For Your Score:**

The paper proposes novel data poisoning techniques for OPE methods that are not well studied by the literature. It helps us understand the effect of such perturbations on the policies selected by the OPE methods.

**Q9 Complying With Reviewing Instructions:**

1: Yes.

---

### Official Review · Reviewer_bzJb · 2022-04-14

**Q2(1) Originality/Novelty:** 3
**Q2(2) Significance/Impact:** 3
**Q2(3) Correctness/Technical Quality:** 3
**Q2(6) Clarity Of Writing:** 4
**Q6 Overall Score:** 7
**Q8 Confidence In Your Score:** 3

**Q1 Summary And Contributions:**

This paper proposes a data poisoning attack, DOPE, for off-policy evaluation methods. The paper proposes a practical attack problem formulation, then apply first-order approximation to the problem, which motivates an effective solving technique for the (approximated) problem. Evaluation on five domains and five learning algorithms show the effectiveness of the attack.

**Q2 Assessment Of The Paper:**

More detailed information regarding each of these aspects is given below:

**Q2(4) Quality Of Experiments (Optional):**

3: Good: The experimental evaluation is adequate, and the results convincingly support the main claims.

**Q2(5) Reproducibility:**

4: Excellent: Key resources (e.g., proofs, code, data) are available and key details (e.g., proof sketches, experimental setup) are comprehensively described for competent researchers to confidently and easily reproduce the main results.

**Q3 Main Strengths:**

- Attack problem formulation and a novel attack technique (to the best of my knowledge, I didn't see other poisoning attacks for off-policy evaluation methods).
- Effective solving techniques via influence functions (especially the greedy method for dealing with sparse L0 constraints).
- Strong empirical results. (Disclaimer: I am not familiar with the off-policy policy evaluation literature and thus unable to evaluate the comprehensiveness of the experimental evaluation. But it looks sufficient for me.)

**Q4 Main Weakness:**

- It appears to be that the paper only considers linear function approximations? I am wondering why such limitations are imposed. I am not familiar with off-policy evaluation literature, however, in offline RL, DNN-based methods are effective which can also be applied to the off-policy evaluation I think? Then, the proposed DOPE attack seems to be capable of attacking these DNN-based off-policy evaluation methods or even general offline RL too. Maybe it is less effective than attacking linear functions due to the impreciseness of first-order approximation. But I think either experiment on DNN-based methods or a discussion on why it can or cannot handle non-linear off-policy evaluation methods could be helpful.

I didn't find other major weaknesses in the paper.

**Q5 Detailed Comments To The Authors:**

The writing quality is generally good. Here are some minor comments on writing:
- Page 2, right column: $\phi(s, a)$ seems to be an $(|\mathcal{A}| \times d)$-dimensional vector from the definition. However, from the latter part of the paragraph, we seem to assume $\phi(s, a)$ to be only $d$ dimensional.
- Page 4, left column, last paragraph: "The loss function $L(\theta, \psi)$ must be twice continuously differentiable and linearly separable with respect to the transitions in $D$." Could you provide some concrete examples of the loss function may be in the appendices so we can see their twice continuity?

**Q7 Justification For Your Score:**

This paper proposes the first attack for off-policy evaluation methods to the best of my knowledge. The attack is based a first-order approximation of a reasonable attack problem formulation, where in the first-order approximation the L0 constraints can be solved by a greedy-based assignment. The empirical evaluation shows the effectiveness of the attack. Although there are some limitations such as limitations to linear function approximations, the merits clearly outweigh the limitations.

**Q9 Complying With Reviewing Instructions:**

1: Yes.

---

### Decision · Program_Chairs · 2022-05-15

**Decision:**

Accept (Oral)

**Comment:**

Meta Review: This work proposes a data poisoning attack for off-policy evaluation (OPE) methods. It also shows that several popular OPE methods are not robust to data poisoning attacks. The reviewers were all positive on this work, recommending strong accept, accept, accept, and weak accept. They praised the generality of the proposed problem and framework.

One main criticism that arose in several of the reviews was a lack of a stronger baseline comparator. In their response, the authors pointed out the challenge of identifying a suitable baseline, given a lack of previous work in data poisoning for off-policy evaluation. In an attempt to provide a suitable baseline, they developed their own comparator, namely a variant of the Fast Gradient Sign Method. They then showed that their main proposed framework outperforms this comparator. In my view, this response adequately addressed this reviewer critique.

The reviewers also made a number of other relatively minor comments and suggestions, and the authors addressed these in their replies.